# Identification of Static Loads in Wharf Mooring Cables Using the Influence Coefficient Method

**DOI:** 10.3390/s25185867

**Published:** 2025-09-19

**Authors:** Jia Zhou, Changshi Xiao, Langxiong Gan, Bo Jiao, Haojie Pan, Haiwen Yuan

**Affiliations:** 1School of Navigation, Wuhan University of Technology, Wuhan 455063, China; 2School of Naval Architecture and Port Engineering, Shandong Jiaotong University, Weihai 264209, China; 3School of Electrical and Information Engineering, Wuhan Institute of Technology, Wuhan 430205, China

**Keywords:** mooring cable load, indirect measurement, load identification, influence coefficient matrix (ICM), strain gauge placement

## Abstract

**Highlights:**

**What are the main findings?**
A novel indirect load measurement framework that integrates the influence coefficient matrix with surface strain data from wharf bollard is proposed to measure mooring cable loads at a wharf;An optimization process is developed for strain gauges placement using a genetic algorithm (GA) approach to improve identification accuracy.

**What is the implication of the main finding?**
The mooring cable loads can be identified with only five strain gauges placed on the surface of the bollard;The accuracy and efficiency of the method are demonstrated through simulation and experimental studies.

**Abstract:**

Directly measuring the mooring cable load while a ship is moored at a wharf poses significant practical challenges. This paper proposes an indirect load measurement method to identify mooring cable static loads based on the Influence Coefficient Matrix (ICM) method. First, a finite element analysis of the bollard is conducted to obtain the full-field strains under each unit load. A solution procedure based on the genetic algorithm (GA) is then implemented to determine the optimal placement and orientation of strain gauges, aiming to improve load identification accuracy. An optimal load coefficient matrix is derived to establish the correlation between cable loads and bollard strains. Subsequently, following the established measured point placement scheme, strain gauges are installed on the bollard surface to capture the strains, enabling inverse identification of mooring cable loads through the measured strains and the pre-established load–strain relationship. A numerical case study validated the feasibility of this method, demonstrating high identification accuracy. Furthermore, experimental verification was conducted to assess its reliability under different conditions. Results confirmed the effectiveness of this indirect approach for mooring cable static loads measurement. The research findings provide a technical framework for real-time monitoring of mooring cable loads.

## 1. Introduction

The mooring safety during ship berthing operations at wharves remains a critical concern in port engineering. As a key load-bearing component connecting the ship to the wharf, mooring cables directly impact personnel safety, ship stability, and the structural integrity of wharf facilities [1]. As modern shipping trends toward larger ships and deeper berths, coupled with complex marine environmental factors and human operational variables, mooring systems frequently face extreme load conditions that exceed design limits. These conditions can lead to major safety accidents, such as cable breakage and bollard structural collapse [2,3,4]. These accidents highlight the urgent need for improved safety monitoring technologies for mooring systems. In this context, the development of high-precision, intelligent, real-time mooring cable load monitoring represents not only a core technical requirement for port operational safety but also a critical infrastructure component for constructing digital twin ports and achieving smart maritime management.

During mooring operations at wharves, the loads on mooring cables exhibit dynamic variations. However, in well-sheltered port areas, these changes occur relatively slowly, allowing the cable loads to be approximated as quasi-static. Consequently, the cable tension measurement may treat such loads as static. Mooring cable tension measurement generally relies on two methods: direct and indirect measurements. Direct methods involve installing sensors on the mooring cables to measure values of strain [5], frequency [6,7,8], acceleration [9,10], or displacement [11], which are then processed to derive the cable tension. However, the mandatory installation of sensors on each mooring cable can interfere with routine berthing and departure operations and introduce safety hazards, thereby limiting their widespread application [12]. In contrast, the indirect measurement method involves installing sensors on the bollard or mooring hook, where structural responses are measured and used to back-calculate the tension in the mooring cables. This approach eliminates the need for frequent sensor disassembly and avoids operational disruptions, making it more suitable for wharf-side tension monitoring. The evolution of sensing technology further supports this method; for instance, emerging passive wireless sensing techniques like Radio-Frequency Identification (RFID) offer promising solutions for long-term strain monitoring with advantages of no external power requirement and ease of installation [13,14,15].

For dangerous cargo ships such as oil carriers, liquefied petroleum gas (LPG) carriers, and liquefied natural gas (LNG) carriers, a mooring monitoring system has been employed to supervise the mooring operations when ships dock [16]. The basic principle of this system is to replace the connection pin between the quick-release mooring hook plate and the ring sleeve with a pin sensor, thus maintaining the connection while indirectly measuring the cable tension. However, due to the high economic cost of the quick-release mooring hook system, this technology is primarily applied in LPG and LNG terminals and is not suitable for docks equipped with conventional bollards [12]. To solve the issue of measuring mooring cable loads at docks that lack quick-release hooks, more attention has been paid to indirect measurement methods based on the response of the bollard structure to calculate cable tension. Compared to substituting entire bollards with mechanically complex quick-release hook systems, deploying sensors on existing bollards and implementing load monitoring through data acquisition and processing units demonstrates potential economic advantages. The core of these methods lies in establishing a mapping relationship between the bollard strain field and cable loads. Mechanical models correlate bollard strains with cable tensions, enabling load inversion through strain measurements at two points [12,17,18,19,20] or three points [21] on the bollard surface. Experimental studies confirm the engineering feasibility of these approaches for safety monitoring and early warning systems [17,18]. Most existing research focuses on the development and verification of strain-to-load models. However, there is little discussion on optimizing sensor placement, which is a crucial factor in improving load inversion accuracy. In current practice, strain gauge placement typically follows empirical rules, such as positioning sensors on the tension side of the bollard at the base section [12]. Machine learning methods—such as Bayesian inference [22] and Gaussian process regression [23]—can enhance the accuracy of load monitoring when processing structural response data. Inadequate theoretical support for strain gauge placement strategies could lead to the omission of optimal measurement points with high load-information density, ultimately degrading the accuracy of load inversion. Theoretical models for optimizing strain gauge placement to address mooring cable load identification challenges have not been systematically explored in the existing literature.

These challenges fundamentally belong to the domain of load identification in structural health monitoring. With the rapid development of sensor placement optimization and load inversion algorithms, various methods have been proposed to improve load identification accuracy [24]. Among these, the Influence Coefficient Matrix (ICM) method has demonstrated exceptional applicability across engineering structures due to its computational efficiency and conceptual clarity [25]. Its framework involves constructing a structural response matrix under unit loads via finite element analysis (FEA) or experimental calibration. Then unknown loads are directly calculated through linear combinations of measured strains and the response matrix. Tim Hunter combined ICM with FEA to develop a commercial software—TrueLoad—which has been used for load identification in mechanical engineering applications, such as suspension components [26] and excavator arms [27]. In marine engineering, ICM has provided solutions for monitoring ice loads on ships [25,28,29,30], steel gate ice-induced loads [31,32,33,34,35], floating offshore wind turbine loads [36,37], as well as for dynamic load identification on ships [38] and bending or torsional moment identification [39,40,41,42,43]. Notably, these studies have achieved indirect load measurement and improved load identification accuracy by combining sensor placement optimization and ICM inversion algorithms, which illustrates the effectiveness of the ICM method in solving load identification problems. This provides a new approach for the indirect measurement of mooring cable loads. However, to the best of the authors’ knowledge, the existing literature on ICM method applications for mooring cable load measurement is limited.

This study proposes a framework for the indirect measurement of mooring cable loads based on the ICM method. First, mechanical analysis and FEA of a bollard is conducted to design unit loads, and the full-field strain under these loads is calculated. To overcome the limitations of traditional empirical sensor placement schemes, the genetic algorithm (GA) is introduced to optimize the strain gauge placement and construct the ICM. Subsequently, the inversion of mooring cable loads is achieved through matrix operations utilizing the ICM and the measured strain data. Compared with existing indirect methods, the innovations include (1) the first systematic application of the ICM method to bollard-based mooring cable load identification, establishing a complete technical path from theoretical modeling to algorithm optimization, and (2) the GA-ICM coupling strategy, which eliminates the reliance on empirical strain gauges placement, providing a quantifiable design basis for sensor placement. This work offers a new technical solution for monitoring mooring cable tension at wharves while contributing to the reliability of existing port mooring safety alert systems.

The structure of this paper is organized as follows. Section 2 introduces the theoretical foundations of the ICM method and GA-based optimization of strain gauge placement; Section 3 describes the mooring cable load identification framework along with preliminary simulation validation; Section 4 validates the proposed load measurement method through a scaled model test; finally, Section 5 summarizes the conclusions.

## 2. Load Identification Method

### 2.1. Influence Coefficient Matrix (ICM) Method

Based on the principles of structural mechanics, when only static effects induced by external loads at a specific instant are considered, and high-frequency dynamic effects are disregarded, the applied load vector ***F*** and the resultant deflection vector ***X*** can be mathematically represented as(1)F=KX
where ***K*** represents the stiffness matrix of the structure. A linear relationship exists between the deflection vector ***X*** and the strain vector ***ε***. This linear relationship holds under the assumptions of small deformations and linear elastic material behavior. Consequently, ***F*** and ***ε*** can be expressed as(2)ε=AF
or(3)$$F=A^{-1}\varepsilon =C\varepsilon$$
where ***A*** is the load coefficient matrix, and ***C***, the influence coefficient matrix, is its inverse.

According to Equation (3), if the strain vector ***ε*** is obtained via physical measurement (e.g., using strain gauges), and the matrix ***C*** is determined, the load vector ***F*** can be calculated. Assuming the structure is subjected to *m* loads and that *n* strain gauges are deployed, the dimensions of the matrices and vectors are defined as ***F*** ∈ R^m×1^, ***ε*** ∈ R^n×1^, ***A*** ∈ R^n×m^, and ***C*** ∈ R^m×n^. Equations (2) and (3) can be expanded as(4)$$\left\{\begin{array}{c} \varepsilon_{1}\\ \varepsilon_{2}\\ \vdots \\ \varepsilon_{n} \end{array}\right\}=\left[\begin{array}{cccc} \begin{array}{c} A_{1,1}\\ A_{2,1}\\ \vdots \\ A_{n,1} \end{array} & \begin{array}{c} A_{1,2}\\ A_{2,2}\\ \vdots \\ A_{n,2} \end{array} & \begin{array}{c} \ldots \\ \ldots \\ \ldots \end{array} & \begin{array}{c} A_{1,m}\\ A_{2,m}\\ \vdots \\ A_{n,m} \end{array} \end{array}\right]\left\{\begin{array}{c} F_{1}\\ F_{2}\\ \vdots \\ F_{m} \end{array}\right\}$$(5)$$\left\{\begin{array}{c} F_{1}\\ F_{2}\\ \vdots \\ F_{m} \end{array}\right\}=\left[\begin{array}{cccc} \begin{array}{c} C_{1,1}\\ C_{2,1}\\ \vdots \\ C_{m,1} \end{array} & \begin{array}{c} C_{1,2}\\ C_{2,2}\\ \vdots \\ C_{m,2} \end{array} & \begin{array}{c} \ldots \\ \ldots \\ \ldots \end{array} & \begin{array}{c} C_{1,n}\\ C_{2,n}\\ \vdots \\ C_{m,n} \end{array} \end{array}\right]\left\{\begin{array}{c} \varepsilon_{1}\\ \varepsilon_{2}\\ \vdots \\ \varepsilon_{n} \end{array}\right\}$$

According to Equation (4), the strain magnitude *ε_n_* at position *n* is(6)$$\varepsilon_{n}=A_{n,1}F_{1}+A_{n,2}F_{2}+\ldots +A_{n,m}F_{m}$$

It is evident that the strain at position *n* results from the superposition of strains induced by m individual loads. Assuming each load acts independently, the loading ***F*** can be represented as an *m × m* matrix. Consequently, Equations (4) and (5) can be expanded as follows:(7)$$\left[\begin{array}{cccc} \begin{array}{c} \varepsilon_{1,1}\\ \varepsilon_{2,1}\\ \vdots \\ \varepsilon_{n,1} \end{array} & \begin{array}{c} \varepsilon_{1,2}\\ \varepsilon_{2,2}\\ \vdots \\ \varepsilon_{n,2} \end{array} & \begin{array}{c} \ldots \\ \ldots \\ \ldots \end{array} & \begin{array}{c} \varepsilon_{1,m}\\ \varepsilon_{2,m}\\ \vdots \\ \varepsilon_{n,m} \end{array} \end{array}\right]=\left[\begin{array}{cccc} \begin{array}{c} A_{1,1}\\ A_{2,1}\\ \vdots \\ A_{n,1} \end{array} & \begin{array}{c} A_{1,2}\\ A_{2,2}\\ \vdots \\ A_{n,2} \end{array} & \begin{array}{c} \ldots \\ \ldots \\ \ldots \end{array} & \begin{array}{c} A_{1,m}\\ A_{2,m}\\ \vdots \\ A_{n,m} \end{array} \end{array}\right]\left[\begin{array}{cccc} \begin{array}{c} F_{1}\\ 0\\ \vdots \\ 0 \end{array} & \begin{array}{c} 0\\ F_{2}\\ \vdots \\ 0 \end{array} & \begin{array}{c} \ldots \\ \ldots \\ \ldots \end{array} & \begin{array}{c} 0\\ 0\\ \vdots \\ F_{m} \end{array} \end{array}\right]$$(8)$$\left[\begin{array}{cccc} \begin{array}{c} F_{1}\\ 0\\ \vdots \\ 0 \end{array} & \begin{array}{c} 0\\ F_{2}\\ \vdots \\ 0 \end{array} & \begin{array}{c} \ldots \\ \ldots \\ \ldots \end{array} & \begin{array}{c} 0\\ 0\\ \vdots \\ F_{m} \end{array} \end{array}\right]=\left[\begin{array}{cccc} \begin{array}{c} C_{1,1}\\ C_{2,1}\\ \vdots \\ C_{m,1} \end{array} & \begin{array}{c} C_{1,2}\\ C_{2,2}\\ \vdots \\ C_{m,2} \end{array} & \begin{array}{c} \ldots \\ \ldots \\ \ldots \end{array} & \begin{array}{c} C_{1,n}\\ C_{2,n}\\ \vdots \\ C_{m,n} \end{array} \end{array}\right]\left[\begin{array}{cccc} \begin{array}{c} \varepsilon_{1,1}\\ \varepsilon_{2,1}\\ \vdots \\ \varepsilon_{n,1} \end{array} & \begin{array}{c} \varepsilon_{1,2}\\ \varepsilon_{2,2}\\ \vdots \\ \varepsilon_{n,2} \end{array} & \begin{array}{c} \ldots \\ \ldots \\ \ldots \end{array} & \begin{array}{c} \varepsilon_{1,m}\\ \varepsilon_{2,m}\\ \vdots \\ \varepsilon_{n,m} \end{array} \end{array}\right]$$
where *ε_n_*_,*m*_ represents the strain caused by the load *F_m_* at position *n*. Therefore, the total strain *ε_n_* can be written as(9)$$\varepsilon_{n}=\varepsilon_{n,1}+\varepsilon_{n,2}+\ldots +\varepsilon_{n,m},$$

Furthermore, when the loads *F*_1_, *F*_2_, …, *F_m_* are all assigned a value of 1, the matrix ***F*** becomes the identity matrix ***I***. Subsequently, the resulting strain matrix ***ε****^I^* can be expressed as(10)$$\varepsilon^{I}=AI=A=\left[\begin{array}{cccc} \begin{array}{c} \varepsilon_{1,1}^{I}\\ \varepsilon_{2,1}^{I}\\ \vdots \\ \varepsilon_{n,1}^{I} \end{array} & \begin{array}{c} \varepsilon_{1,2}^{I}\\ \varepsilon_{2,2}^{I}\\ \vdots \\ \varepsilon_{n,1}^{I} \end{array} & \begin{array}{c} \ldots \\ \ldots \\ \ldots \end{array} & \begin{array}{c} \varepsilon_{1,m}^{I}\\ \varepsilon_{2,m}^{I}\\ \vdots \\ \varepsilon_{n,m}^{I} \end{array} \end{array}\right]$$(11)$$I=C\varepsilon^{I}$$

In general, matrices ***A*** and ***ε****^I^* are not square matrices since *m* and *n* are not necessarily equal. Therefore, Equation (11) may require a pseudo-inverse relationship to determine the value of the matrix ***C***.(12)$$C={\left[{\left(\varepsilon^{I}\right)}^{T}\varepsilon^{I}\right]}^{-1}{\left(\varepsilon^{I}\right)}^{T}$$

Substituting Equation (12) into (3) yields the following result for load identification:(13)$$\hat{F}=\left({\left[{\left(\varepsilon^{I}\right)}^{T}\varepsilon^{I}\right]}^{-1}{\left(\varepsilon^{I}\right)}^{T}\right)\hat{\varepsilon }$$
where $\hat{\varepsilon }$ is the measured strain matrix induced by the applied load matrix $\hat{F}$. It is important to note that the strain matrix ***ε****^I^* varies depending on the selected measurement positions and orientations. Consequently, the influence coefficient matrix ***C*** will also change accordingly. The measurement positions and orientations for $\hat{\varepsilon }$ must be consistent with those used to construct ***ε^I^*** and ***C***.

### 2.2. Optimization of Strain Gauge Placement

The construction of the influence coefficient matrix ***C*** is dependent on the placement of strain gauges. A pivotal research question involves determining how to strategically position a minimal number of sensors at structurally critical locations to maximize the comprehensiveness of acquired load information. Current methodologies for sensor placement optimization include the effective independence (EI) method [44] and heuristic algorithms such as the genetic algorithm (GA) [45], particle swarm optimization (PSO) method [46], artificial fish swarm algorithm [47], and other intelligent algorithms [48,49]. Among these, the GA is a population-based evolutionary computing technique. This method demonstrates exceptional generalization capability due to its problem-agnostic nature, making it particularly effective for sensor placement optimization [50]. To enhance the robustness and stability of C, this study employs the GA method to optimize strain gauge placement. The algorithm follows a workflow comprising these steps: (1) population initialization; (2) fitness evaluation of individuals; (3) iterative execution of selection, crossover, and mutation operations until termination criteria are satisfied; and (4) identification of the optimal layout.

#### 2.2.1. Individual Code

In this research, strain gauges are conceptualized as genetic elements, with each potential placement configuration representing an individual. For a selected number of measurement points *n*, the position of the strain gauge *i* is denoted as *p_i_*, and its mounting direction is represented by γ_i_; the encoding of the mounting placement scheme *j* can be expressed as(14)$$Individual^{j}=\left\{p_{1}^{j},p_{2}^{j},\ldots ,p_{i}^{j},\ldots p_{n}^{j},\gamma_{1}^{j},\gamma_{2}^{j},\ldots ,\gamma_{i}^{j},\ldots \gamma_{n}^{j}\right\}$$

The initialization of the population ***P*_0_** is performed by generating a predefined number of individuals via stochastic sampling.

#### 2.2.2. Fitness Function

The matrix inverse can be calculated by dividing the adjoint matrix (transposed cofactor matrix). Therefore, the inverse operation in Equation (12) can be expressed as(15)$${\left[{\left(\varepsilon^{I}\right)}^{T}\varepsilon^{I}\right]}^{-1}=\frac{adj\left({\left(\varepsilon^{I}\right)}^{T}\varepsilon^{I}\right)}{\left|{\left(\varepsilon^{I}\right)}^{T}\varepsilon^{I}\right|}$$

The stability of Equation (15) affects the stability of matrix ***C***. Thus, the fitness function of individuals is defined as(16)$$\left|{\left(\varepsilon^{I}\right)}^{T}\varepsilon^{I}\right|\to \max$$
where ***ε^I^*** is composed of the strains of strain gauge placement scheme *j* when each unit load is individually applied to the structure.

#### 2.2.3. Genetic Operations

The selection operator is applied to the population, where individuals are evaluated based on their fitness, and the fittest individuals are preserved for the next generation. The crossover operator denotes the exchange of genetic segments between two parent individuals to generate offspring, serving as a core mechanism in genetic algorithms to explore the search space and combine advantageous traits from diverse solutions. The mutation operator introduces diversity by stochastically altering gene values at specific positions within individual chromosomes, thereby enabling the discovery of novel solutions. Following these operations, the population ***P_t_*** evolves into the subsequent generation ***P_t_*_1_**.

#### 2.2.4. Termination Conditions

When the iteration count reaches the predefined maximum *T*, the algorithm terminates and returns the individual with the highest fitness value observed throughout the optimization process as the optimal solution.

## 3. Indirect Load Measurement for Mooring Cable

### 3.1. Load Matrix of Mooring Cable

Common types of wharf bollards include single-cross, double-cross, upright, inclined, and horn-shaped configurations. To prevent cable slippage, bollards are generally equipped with caps having larger diameters than their main bodies. During ship berthing operations, environmental forces (e.g., wind, waves, and currents) can cause vessel displacement toward the wharf’s periphery, leading to cable-induced tensile forces on mooring bollards.

This study analyzes the mechanical behavior of a mooring bollard under cable loading (Figure 1). In static equilibrium, the cable wrapped around the bollard establishes contact along a circumferential arc of the cross-sectional profile of the bollard. Frictional effects at the cable–bollard interface are assumed negligible, with only tensile and flexural forces considered in the model. Consequently, the distributed load is analytically approximated as a concentrated force *F* applied at the critical contact point *S*. The force *F* can be mathematically decomposed into an axial force *F_a_* and a radial force Fr. The axial force *F_a_* induces tensile deformation along the bollard’s longitudinal axis while simultaneously establishing a bending moment that generates non-uniform stress distributions across transverse cross-sectional planes. The radial *F_r_* predominantly induces flexural deformation through transverse loading. The strain on the surface of the bollard can be described as the superposition of axial tensile strain due to *F_a_* and flexural strain due to *F_r_*.

Taking into account the actual situation, precise determination of the force point on the edge of the bollard is challenging because the direction of the cable changes with the movement of the ship. Consequently, the force point is shifted from *S* to the axial center *O* of the bollard. For the axial force *F_a_*, a force couple *M* = *F_a_ × R* is introduced at *O* via the parallel axis theorem. The radial force *F_r_* is directly shifted to point *O* under the assumption of negligible bollard section deflection. The resultant force *F* and couple *M* at *O* are resolved into Cartesian components *F_x_*, *F_y_*, *F_z_*, *M_x_*, and *M_y_*, as illustrated in Figure 1. Let *ɸ* denote the angle between *F_a_* and the positive *Z*-axis and *θ* represent the angle between *F_r_* and the positive *Y*-axis (clockwise positive). The following relationships hold:(17)$$\left\{\begin{array}{l} F_{x}=F_{r}\sin\theta =F\sin ɸ\sin\theta \\ F_{y}=F_{r}\cos\theta =F\sin ɸ\cos\theta \\ F_{z}=F_{a}=F\cos ɸ\\ M_{x}=M\sin\theta =F_{a}R\sin\theta \\ M_{y}=M\cos\theta =F_{a}R\cos\theta \end{array}\right.$$

For the force *F*, there is(18)$$F=\sqrt{F_{r}^{2}+F_{a}^{2}}=\sqrt{F_{x}^{2}+F_{y}^{2}+F_{z}^{2}}$$

Based on the preceding analysis, the mooring cable load identification problem is reduced to identifying the three concentrated forces *F_x_*, *F_y_*, and *F_z_* acting at the axial center *O* of the bollard. When employing the method of the influence coefficient matrix, the effect of moment *M* should be considered when constructing the load matrix. Consequently, the load matrix is a 5 × 5 matrix, as expressed in Equation (19):(19)$$F=\left[\begin{array}{ccccc} F_{x} & 0 & 0 & 0 & 0\\ 0 & F_{y} & 0 & 0 & 0\\ 0 & 0 & F_{z} & 0 & 0\\ 0 & 0 & 0 & M_{x} & 0\\ 0 & 0 & 0 & 0 & M_{y} \end{array}\right]$$

### 3.2. Full-Field Strain Response of Mooring Bollard

To construct the influence coefficient matrix using strain data from strategically positioned measurement points on a mooring bollard, it is imperative to initially acquire the full-field strain response of the mooring bollard structure under various unit loads. These strain distributions are determined via FEA. In this study, a specific bollard depicted in Figure 2 serves as the subject of a case study.

The bollard features a nominal diameter of 108 mm, wall thickness of 4 mm, height of 200 mm, and a unilateral cap extension with one side of the bollard cap extending outward. The bollard is constructed from steel with a Young’s modulus of 206 GPa and Poisson’s ratio of 0.3.

Furthermore, the base of the mooring bollard is constrained by fully fixed boundary conditions. The force and moment act at the axial center of the bollard top. To achieve realistic strain distributions under axial bending moments, a rigid coupling constraint is implemented between the top axial center point and the perimeter of the bollard’s circular cross-section. The cylindrical portion of the model, excluding the bollard cap, is divided into 2400 elements, with meshing performed using four-node thin shell elements for meshing (chosen for computational efficiency given the structure’s thin-walled nature). The finite element model is illustrated in Figure 3.

To ensure comprehensive representation of each unit load’s influence within the coefficient matrix, the selection of the unit load magnitudes is critical for generating measurable and distinguishable strain responses. In this paper, the directions of the unit load are shown in Figure 3. The values are defined as *F_x_* = *F_y_* = *F_z_* = 10 kN; *M_x_* = *M_y_* = −500 kN·mm, which means the unit load matrix is(20)$$I=\left[\begin{array}{ccccc} 10 & 0 & 0 & 0 & 0\\ 0 & 10 & 0 & 0 & 0\\ 0 & 0 & 10 & 0 & 0\\ 0 & 0 & 0 & -500 & 0\\ 0 & 0 & 0 & 0 & -500 \end{array}\right]$$

Therefore, it is necessary to use the finite element simulation technique to calculate the full-field strain information of the mooring bollard under the five unit load conditions. This strain information can be articulated by the three planar strain components *ε_x_*, *ε_y_*, and *γ_xy_* of the shell elements. The strain results of each condition are shown in Figure 4, Figure 5, Figure 6, Figure 7 and Figure 8.

By using the transformation Equation (21) for plane strain, the strain magnitude in the direction *ψ* of the strain gauge can be calculated.(21)$$\varepsilon_{\psi }=\varepsilon_{x}\cos^{2}\psi +\varepsilon_{y}\sin^{2}\psi +\gamma_{xy}\sin\psi \cos\psi$$

### 3.3. Strain Gauge Placement and Influence Coefficient Matrix

Given the inherent simplifications in the finite element model’s force and boundary condition assumptions, significant discrepancies may arise between simulated and experimentally measured strains, particularly in proximal regions of load application and boundary constraints. Consequently, these strain values should be excluded from the pool of potential locations for strain gauge placement. Through the GA search methodology, as detailed in Section 2.2, optimal sensor positions and orientations were determined by systematically screening valid candidate elements. Typically, under the assumption of unimpeded strain gauge integrity and continuous data acquisition, the number of strain gauges required for load identification equals the number of loads. This study employed five measurement points for cable load identification, corresponding to the number of unit loads (five). The GA-based methodology and the parameters used for optimizing strain gauge placement are detailed in Table 1.

Candidate regions were defined by excluding areas within 30 mm of the fixed boundary and 75 mm of the top load application zone. Based on the strain values of these candidate regional units, an initial population is created and iteratively computed through genetic operators. Figure 9 shows the variation curve of fitness values. As can be seen from the figure, the fitness value tends to stabilize after 150 iterations.

Following the aforementioned procedure, the finalized strain gauge configuration (Figure 10) positions gauges G1–G3 at 37.5 mm above the bollard base and G4–G5 at 122.5 mm. The orientations of the five strain gauges are all close to 90°; that is, along the axis of the bollard. Referring to the strain contour plots of the finite element results, it can be observed that all measuring points are situated within strain-sensitive areas. Subsequently, the influence coefficient matrix can be derived using the FEA strain results of these five locations under each unit load.

### 3.4. Simulation Verification

To verify the effectiveness of the identification method proposed in this paper, this study identified mooring cable loads of 5.5 kN in various directions. Specifically, angles *ɸ* and *θ* (Figure 1) represent these different acting directions, resulting in nine distinct conditions. The load components for each case are shown in Table 2.

Strain data were acquired through finite element analysis, where the mesh discretization emulates physical strain gauge placements. Analogous to the application of unit load, the cable load is applied at the center of the bollard section with force and moment components.

By leveraging FEA, simulated strain values ${\hat{\varepsilon }}_{s}$ can be obtained at each measuring point. By multiplying ${\hat{\varepsilon }}_{s}$ with the influence coefficient matrix ***C***, the load components in each direction can be calculated. The resultant mooring cable loads were then derived using Equation (18). As illustrated in Figure 11, Figure 12 and Figure 13, the FEA-based identification results exhibit exceptional agreement with ground-truth loads in both magnitude and angular orientation. The accuracy of load identification is very high, and the errors of results under different conditions are all less than 0.1%. This demonstrates that the mooring cable load identification method described in this article is effective.

## 4. Experimental Verification

To validate the applicability of the proposed method in measuring the mooring cable load in the scenario depicted in Figure 1, a bespoke bollard and a customized tension-providing frame were designed and constructed.

### 4.1. Description of the Experiment

The experimental setup and physical installation configuration are illustrated in Figure 14. The bollard assembly consists of three primary components: a base, a cylindrical body, and a cap. The cylindrical body is fabricated from Q235-grade seamless steel tubing with an outer diameter of 108 mm and a wall thickness of 4 mm, and it is welded to 12 mm thick steel plates at both ends to form the cap and base. The load application system includes a reinforced foundation, a modular support frame constructed from square steel tubing, and a winch platform. The support frame integrates adjustable pulley assemblies mounted on Component A to set the tensile angle *ɸ* (see Table 2). A manual winch with a capacity of 11.8 kN (2600 lb) is installed on the platform to apply tensile forces. The entire system is anchored to the ground with concrete and secured by multiple anchor bolts. The bollard and load application system were installed with a 2 m separation distance and an elevation of 0.2 m above ground, ensuring horizontal alignment between the lower edge of the bollard cap and the baseline of Pulley 1.

A synthetic fiber rope with a breaking strength of 10 kN was used as the laboratory-scale simulated mooring cable. To accurately measure the tension applied to the cable, a 20 kN capacity tension meter was integrated into the system and connected in series between the cable and the winch wire rope. Uniaxial strain gauges (type BE120-3AA, resistance 120 Ω ± 0.5%, gauge factor 2.1) were bonded to predefined locations on the bollard’s exterior surface for high-precision strain monitoring. The strain data were transmitted to a data acquisition system via shielded twisted-pair cables, which, for the purposes of this initial prototype validation, were sufficient to ensure noise immunity and real-time analysis in a controlled environment. Notably, high-sensitivity sensors should be prioritized in practical implementations to prevent signal loss under low-magnitude loads. Moreover, metallic strain gauges inherently exhibit zero-drift and creep phenomena during operation. For long-term monitoring, fiber-optic strain gauges are strongly recommended due to their superior metrological stability.

Owing to the rigid anchoring of both the bollard and the tension-providing frame to the ground, the force application angle *θ* (defined in Table 2) was fixed at 0°. To simulate lateral loading conditions at 30° and 60°, a strategic experimental protocol was adopted: rather than altering the cable’s force direction, the positioning of the strain gauge measuring points was rotated to align with the desired angles. This operational strategy is schematically represented in Figure 15.

### 4.2. FEA Model Updating

The experiments were performed under the conditions described in Section 4.1, utilizing a data sampling frequency of 2 Hz. When the tension force was 5.5 kN, experimental strain readings from the 15 predefined measurement points at different tensile angles (*ɸ*) are presented in Figure 16.

A systematic discrepancy was observed between experimental strain values and FEA results at certain locations, particularly when strain gauge readings fell below 20 microstrain (με), where the signal-to-noise ratio may compromise measurement accuracy. For instance, the maximum error of the comparison results of G5_2 at *ɸ* = 90° is approximately 14%, as shown in Figure 17. The parameters employed in finite element (FE) computational models are often empirically determined, inherently introducing deviations from the strain responses observed in actual structures. This can significantly skew the outcomes of load identification. Therefore, it is crucial to refine the simulation models by calibrating their parameters against experimental data from structural tests. This process, referred to as FE model updating, aims to minimize the discrepancies between simulation analyses and experimental measurements.

In comprehensive reviews of finite element (FE) model updating techniques conducted by scholars [51,52,53], the methodologies are generally classified into two primary categories: direct methods [54] and parameter-based methods [55]. The direct methods operate through direct modification of the structural mass and stiffness matrices. However, the updated matrices frequently exhibit compromised band sparsity characteristics, resulting in ambiguous physical interpretability [56]. Consequently, the practical applicability of this methodology is significantly constrained. In contrast, the parameter-based approach offers distinct advantages due to its unambiguous physical interpretation and capacity to preserve the banded sparse symmetry of the mass and stiffness matrices post correction. This study adopts parameter-based model updating to refine the FE model. Grounded in this methodology and contextualized to the experimental system, the following optimization framework is formulated:(22)$$\begin{array}{ll} \min & f(D,t)=\sqrt{\frac{1}{n}\sum_{i=1}^{n}{\left(\varepsilon_{i}^{a}-\varepsilon_{i}^{e}\right)}^{2}}\\ s.t. & D_{\min}<D<D_{\max}\\ & t_{\min}<t<t_{\max} \end{array}$$
where *D* and *t* denote the outer diameter and wall thickness of the bollard, respectively. Accounting for manufacturing tolerances, both parameters are treated as variables to be modified. According to the tolerance grade for hot-finished seamless steel tubes, the permissible deviation for the outer diameter of the test bollard is ±0.4 mm, and that for the wall thickness is ±0.2 mm. Consequently, the bounds for *D* are set at *D_min_* = 107.5 mm and *D_max_* = 108.5 mm, while those for *t* are *t_min_* = 3.80 mm and *t_max_* = 4.20 mm. *f*(*D*, *t*) represents the objective to be updated, which is equivalent to the mean square error (MSE) of strain values at all measurement points. $\varepsilon_{i}^{a}$ and $\varepsilon_{i}^{e}$ denote the strain values at measurement point i obtained from finite element analysis and experimental testing, respectively. n is the total number of measurement points.

This study calculates the strain values at measurement points for 25 combinations of outer diameter and wall thickness, with the outer diameter varying within the range of [107.6, 107.8, 108.0, 108.2, 108.4] mm and the wall thickness within [3.8, 3.9, 4.0, 4.1, 4.2] mm. The strain values of the 15 measurement points from the three orientations with 5.5 kN tension are substituted into Equation (20) for model-updating calculations. The optimal combination of outer diameter and wall thickness, which minimizes the objective function value, is identified as *D* = 108.0 mm and *t* = 4.10 mm. Table 3 summarizes the strain values and errors at each measurement point before and after the finite element model updating.

As evident from the results presented in Table 3, although the error variations at individual measurement points differ after the finite element (FE) model updating, the reduction in the objective function value indicates a decrease in the overall error between the FE calculations and experimental results. Consequently, utilizing the FE model with parameters *D* = 108.0 mm and *t* = 4.10 mm, unit loads are applied following the methodology outlined in Section 3.2 to obtain the full-field strain response of the bollard after model updating.

### 4.3. Experimental Results Analysis

By utilizing the aforementioned experimental methodologies, the measured surface strain ${\hat{\varepsilon }}_{e}$ of the test bollard was obtained under various tension conditions listed in Table 2. Directional load components were computed using Equation (13), with the experimental results for load magnitudes and angular orientations detailed in Figure 11 and Figure 13, respectively. The final identification results of the mooring cable load were calculated using Equation (18) and presented in Figure 12. In these figures, load identification values from finite element simulation and experiment are presented as two sets of scatter points, with their corresponding deviations from the equality lines indicated by dashed error bars.

Analysis of the experimental results revealed the following patterns:

(1) Identification of load components. As shown in Figure 11, the maximum errors associated with the five load components varied. The identification accuracy for the horizontal loads (*F_x_* and *F_y_*) was higher than that for the vertical load (*F_z_*) and the bending moments (*M_x_* and *M_y_*). These discrepancies can be primarily attributed to differences in load application methods. In the finite element analysis, forces and equivalent moments were idealized as applied at the axial center of the bollard. During practical experiments, however, the load was introduced eccentrically at the edge, leading to variations in structural strain responses. Since the horizontal load components act through the top center, the strain response differences between the two loading modes are relatively small. In contrast, vertical loads and the moments induced by them elicit more pronounced changes in strain response. Since the location and direction of the mooring cable load are unknown prior to identification, the unit load in FEA can only be applied using the center loading method when constructing the influence coefficient matrix. Experimentally, however, structural strains were measured under edge-loading conditions. This inconsistency in loading configuration contributes to the observed identification errors.

(2) Identification of load direction. As illustrated in Figure 12, the identification errors for both the horizontal angle (*θ*) and vertical angle (*ɸ*) were within ±10%. The errors did not exhibit a consistent trend and were largely stochastic across the tested cases. This randomness may be attributed to the two main causes: deviations in load direction and measurement inaccuracies in the strain gauges. Although the tension meter accurately controlled the magnitude of the cable load during testing, the direction was not mechanically regulated and relied on manual alignment, introducing directional deviation. Consequently, the identified load direction and its components also deviate from the required values. The angular setting errors can be significantly reduced through automatic alignment systems. Additionally, strain gauges inherently possess measurement errors. Positioning strain gauges accurately on a cylindrical bollard was challenging, often resulting in misalignment between the actual position and the intended measurement points. Despite implementing finite element updating to mitigate these errors, complete elimination was not feasible.

(3) Identification of mooring cable load. Although the maximum errors in the identification of individual load components ranged from 10% to 20%, the error in the identified mooring cable load (*F*) did not exceed 5%. This can be attributed to the fact that the cable tension is a result value derived from multiple components (see Equation (18)), and the maximum errors of each component did not occur at the same time.

Overall, the experimental results align closely with the true values, indicating that the method presented in this paper provides satisfactory accuracy for load identification in mooring cables.

### 4.4. Extended Validation Experiments

To further verify the general applicability of the proposed methodology, additional validation experiments were conducted under diverse loading conditions. The variation of the mooring force *F* with varying horizontal angle *θ* and vertical angle *ɸ* was simulated in the test, and the loading condition is detailed in Figure 18.

Surface strain measurements at designated locations of the test bollard were recorded for each loading condition and subsequently processed through the derived load identification equations. The identification results are presented in Figure 19.

Error analysis reveals that the maximum relative error in load magnitude and direction remains below 10%, which validates the accuracy and reliability of the proposed method.

## 5. Conclusions

This study proposes a mooring cable load identification method for wharf applications, leveraging the ICM method to inversely calculate loads through finite element analysis FEA data and strain measurements. Specifically, the loads exerted by the mooring cable on the bollard are approximated as concentrated forces and moments in various directions. Firstly, the finite element analysis is used to calculate the full-field strain of the bollard under each unit load. Subsequently, the positions and orientations of strain gauges are optimized using the GA method, enabling the construction of the influence coefficient matrix. Ultimately, the collected strain data are substituted into the load inversion equation to identify the load.

The results show that it is possible to estimate the mooring cable load using only five strain gauges, achieving a relative error below 10% compared to actual values. This level of accuracy is satisfactory for engineering applications, validating the feasibility of the approach. However, the accuracy of the method depends on the influence coefficient matrix as well as the measurement errors. The influence coefficient matrix is mainly affected by three factors: (1) the simplification of mooring cable loads, (2) the precision of the bollard’s FEA, and (3) the choice of strain gauge positions and orientations. The quantitative impacts of these factors on identification accuracy require further methodical examination in subsequent research. Notably, while transferring the load application point from the bollard’s surface to its axis introduces deviations from reality, the obtained load identification results remain acceptable.

This study establishes a framework and demonstrates the promising applicability of the Influence Coefficient Method (ICM) for identifying mooring cable loads in wharves. However, it is important to note that the current work focuses exclusively on the static load identification of a single bollard with a single cable, which diverges from the more complex dynamic load conditions typical of real-world engineering scenarios involving multiple bollards and cables. To bridge this gap, future studies should explore the extension of the ICM to dynamic and multi-cable systems, with an emphasis on validating its practical feasibility and reliability under realistic conditions.

## Figures and Tables

**Figure 1 sensors-25-05867-f001:**
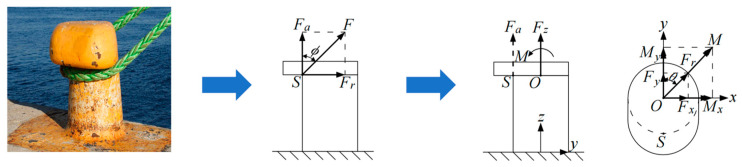
Force analysis of a mooring bollard.

**Figure 2 sensors-25-05867-f002:**
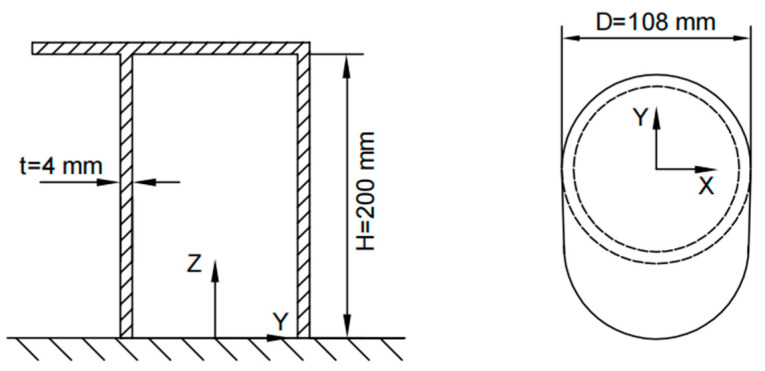
The dimension of the bollard.

**Figure 3 sensors-25-05867-f003:**
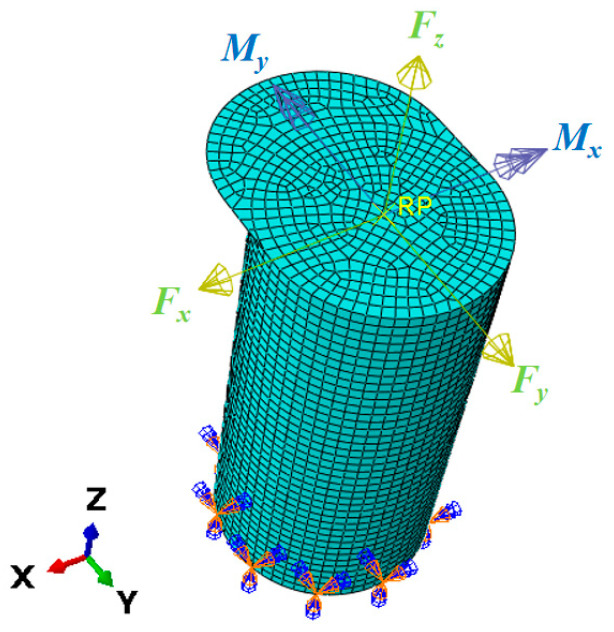
FEA model of the bollard.

**Figure 4 sensors-25-05867-f004:**
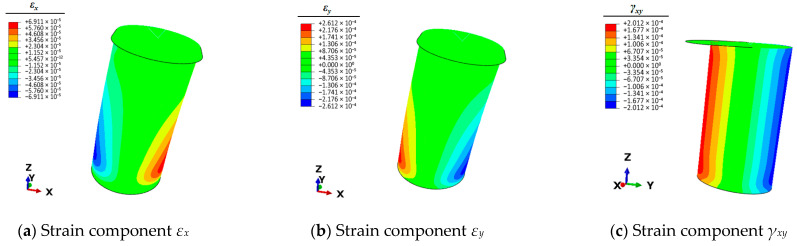
Contour plots of strains for bollard corresponding to unit load *F_x_*.

**Figure 5 sensors-25-05867-f005:**
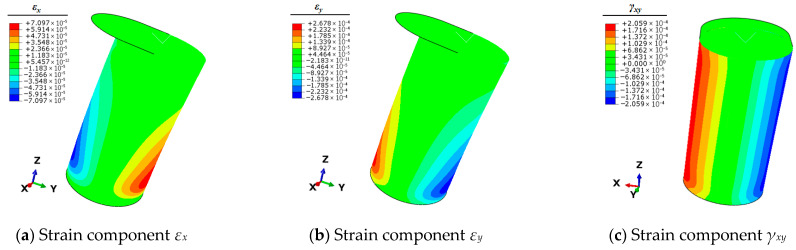
Contour plots of strains for bollard corresponding to unit load *F_y._*

**Figure 6 sensors-25-05867-f006:**
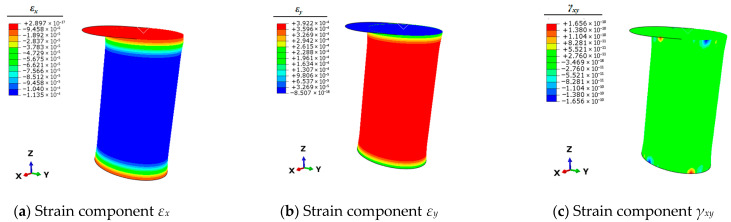
Contour plots of strains for bollard corresponding to unit load *F_z_*.

**Figure 7 sensors-25-05867-f007:**
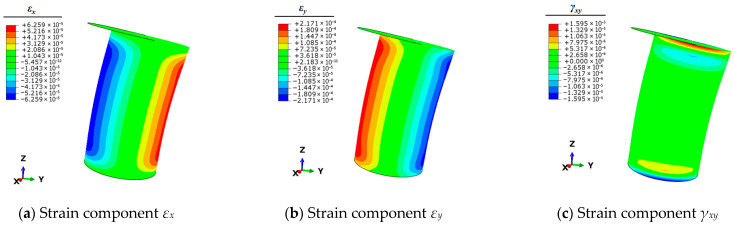
Contour plots of strains for bollard corresponding to unit load *M_x_*.

**Figure 8 sensors-25-05867-f008:**
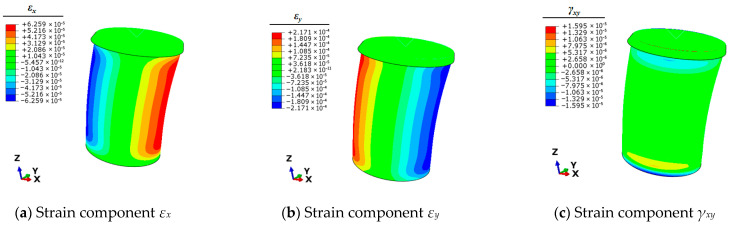
Contour plots of strains for bollard corresponding to unit load *M_y_*.

**Figure 9 sensors-25-05867-f009:**
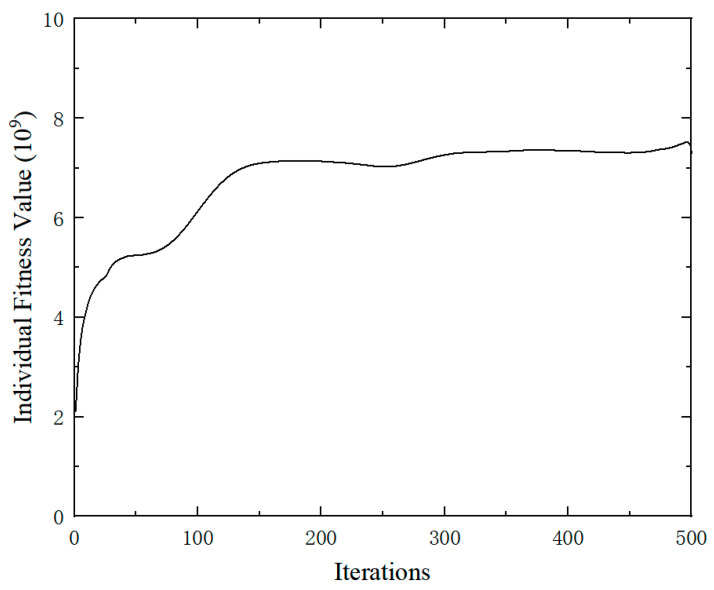
The fitness values curve.

**Figure 10 sensors-25-05867-f010:**
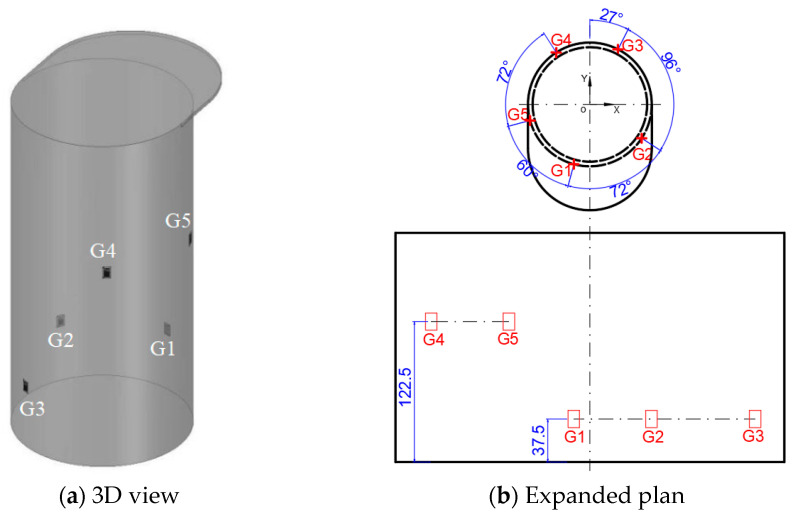
The optimum strain gauge placement.

**Figure 11 sensors-25-05867-f011:**
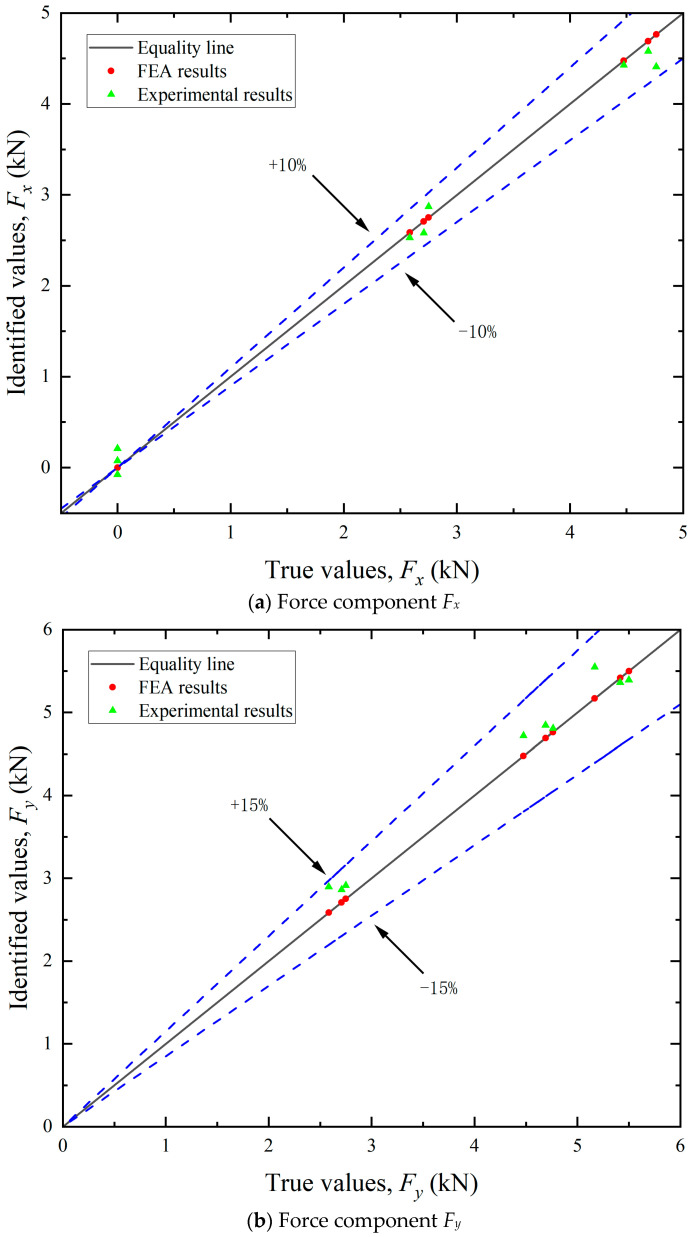
Identification results of the load components.

**Figure 12 sensors-25-05867-f012:**
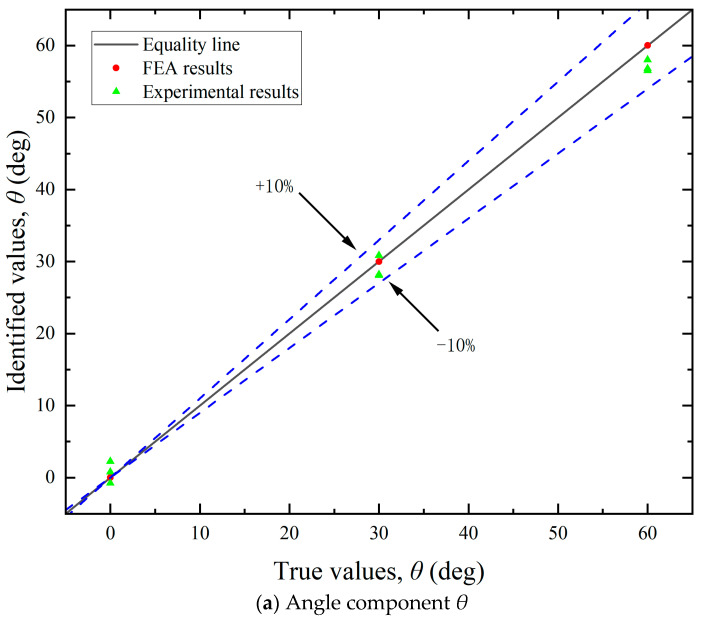
Identification results of the angle components.

**Figure 13 sensors-25-05867-f013:**
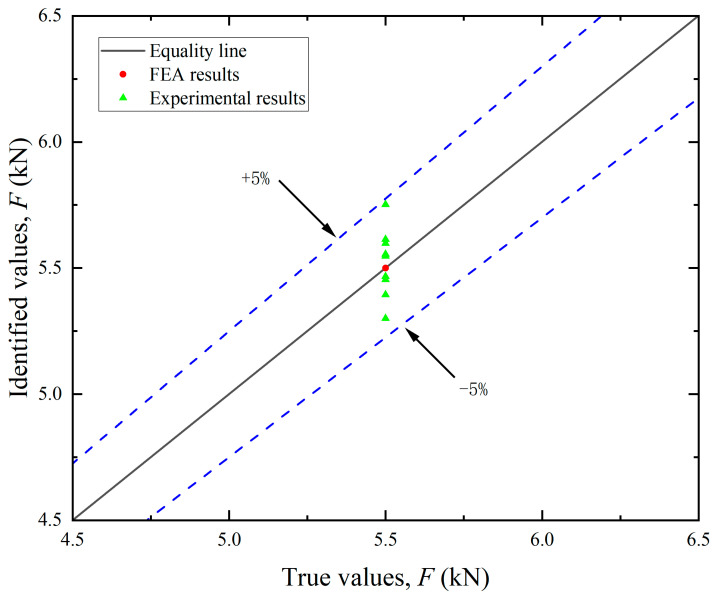
Identification results of the mooring cable load *F*.

**Figure 14 sensors-25-05867-f014:**
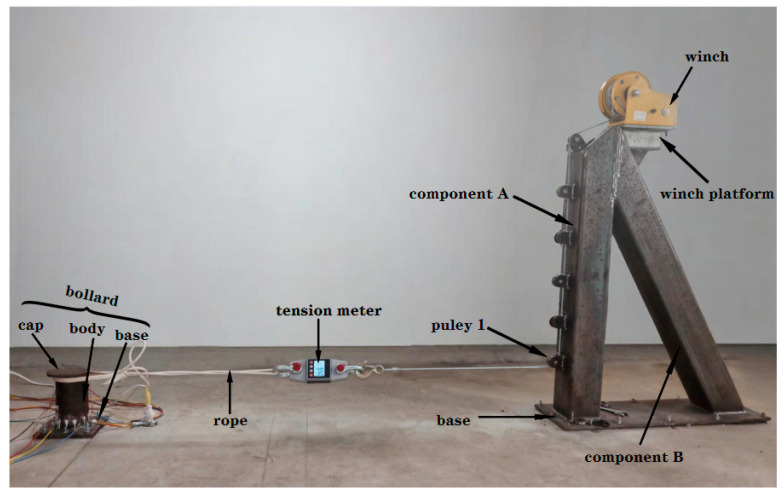
The experimental equipment.

**Figure 15 sensors-25-05867-f015:**
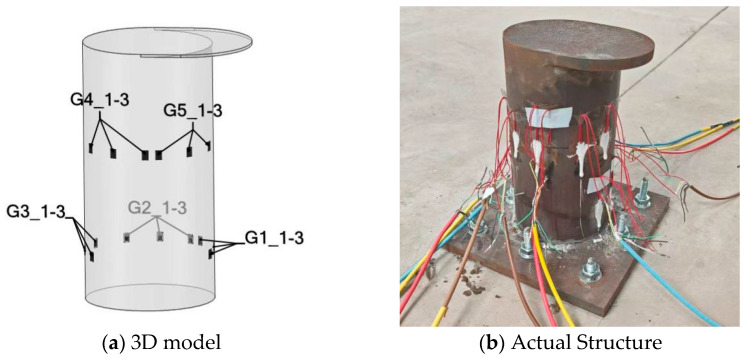
Strain gauge placement.

**Figure 16 sensors-25-05867-f016:**
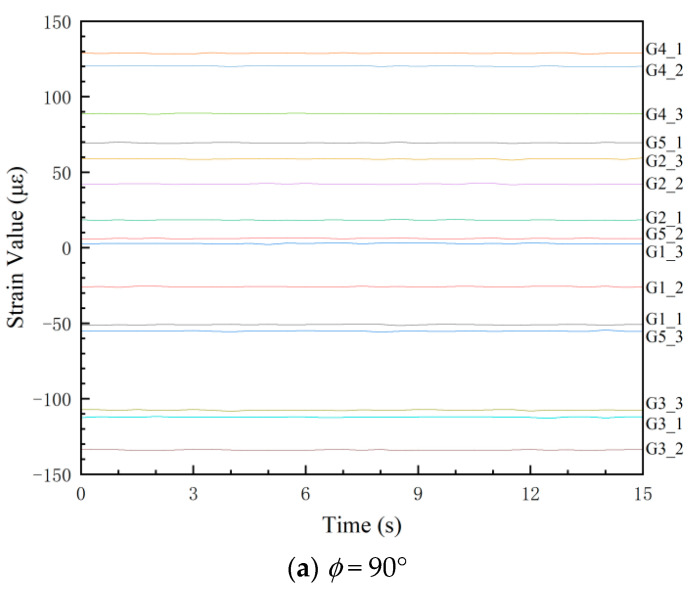
Experimental strain values.

**Figure 17 sensors-25-05867-f017:**
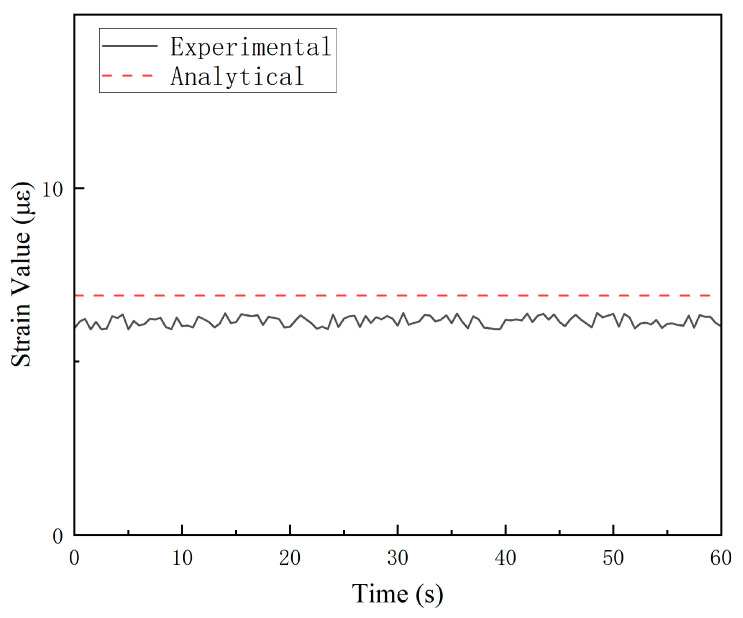
Comparison between experimental and analytical values of strain gauge G5_2.

**Figure 18 sensors-25-05867-f018:**
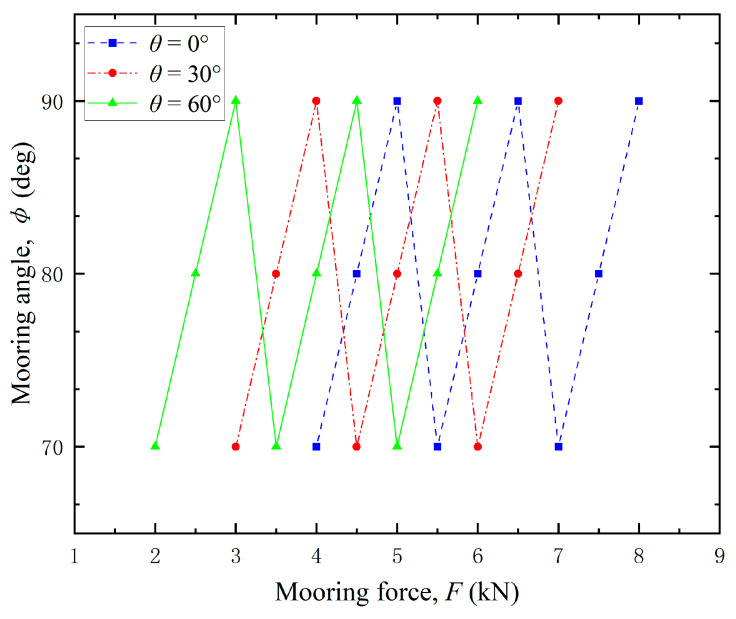
Loading condition.

**Figure 19 sensors-25-05867-f019:**
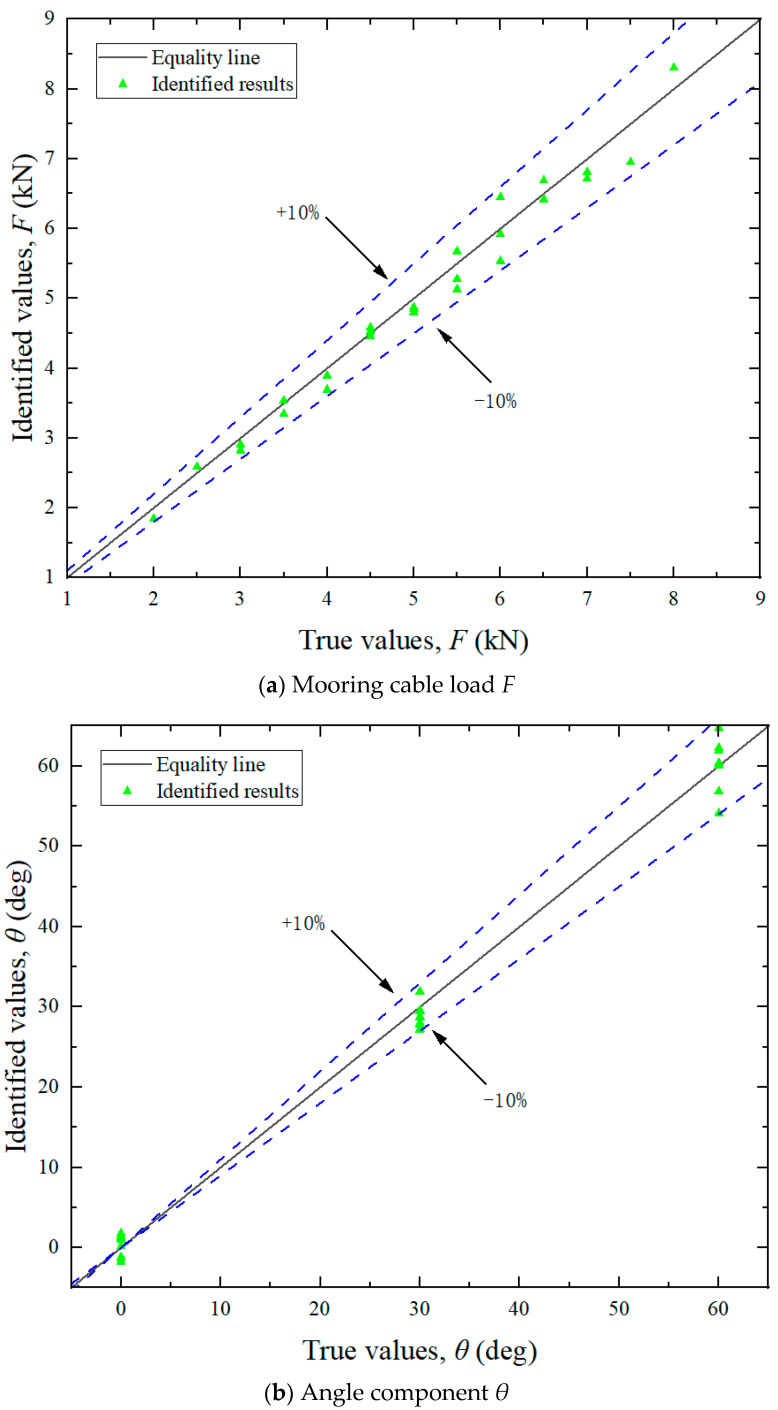
Identification results.

**Table 1 sensors-25-05867-t001:** Pseudo code of the GA-based strain gauges location optimization method.

Algorithm GA-Based Strain Gauges Location Optimization Method
Input:
Number of strain gauges: *n* (=5)
Number of loads: *m* (=5)
Population size: *N* (=200)
Iterations: *T* (=500)
Crossover probability: *P_c_* (=0.7)
Mutation probability: *P_m_* (=0.1)
Candidate strain sets under various unit load: $\varepsilon_{F_{x}},\varepsilon_{F_{y}},\varepsilon_{F_{z}},\varepsilon_{M_{x}},\varepsilon_{M_{y}}$(which are from FEA results.)
Allowable orientation of strain gauges: 0–180°
Output:
Global optimal solution: $x^{\mathrm{best}}$
Influence coefficient matrix: *C*
Procedure:
1 Population initialization using random sample, individuals are constructed as per Equation (14).
2 while Iter < T
3 for each individual *x_j_*
4 1 Calculate the strain of element *n* at angle *ψ* direction under unit load *F_m_*, $\varepsilon_{m,n}$ as per Equation (21).
5 2 Construct the strain matrix caused by unit loads, $\varepsilon^{I}$ as per Equation (10).
6 3 Calculate individual fitness value and select the best individual. Fitness values are calculated as per Equation (16).
7 end for
8 Iter = Iter + 1
9 end while
10 Output global optimal solution $x^{\mathrm{best}}=\left\{p_{1}^{best},p_{2}^{best},p_{3}^{best},p_{4}^{best},p_{5}^{best},\psi_{1}^{best},\psi_{2}^{best},\psi_{3}^{best},\psi_{4}^{best},\psi_{5}^{best}\right\}$ (*p_i_* represents the element number, *ψ_i_* represents the orientation of strain gauge)
11 Calculate influence coefficient matrix *C* as per Equation (12).

**Table 2 sensors-25-05867-t002:** Load components of mooring cable (5.5 kN) in different directions.

Case No.	*ɸ*	*θ*	*F_a_*	*F_r_*	*F_x_*	*F_y_*	*F_z_*	*M_x_*	*M_y_*
	deg	deg	kN	kN	kN	kN	kN	kN·mm	kN·mm
1	70	0	1.881	5.168	0.000	5.168	1.881	−101.580	0.000
2	80	0	0.955	5.416	0.000	5.416	0.955	−51.574	0.000
3	90	0	0.000	5.500	0.000	5.500	0.000	0.000	0.000
4	70	30	1.881	5.168	2.584	4.476	1.881	−87.971	50.790
5	80	30	0.955	5.416	2.708	4.691	0.955	−44.664	25.787
6	90	30	0.000	5.500	2.750	4.763	0.000	0.000	0.000
7	70	60	1.881	5.168	4.476	2.584	1.881	−50.790	87.971
8	80	60	0.955	5.416	4.691	2.708	0.955	−25.787	44.664
9	90	60	0.000	5.500	4.763	2.750	0.000	0.000	0.000

**Table 3 sensors-25-05867-t003:** Strain values and errors at each measurement point.

	Strain Values (*ɸ* = 90°)				Strain Values (*ɸ =* 80°)				Strain Values (*ɸ* = 70°)			
Points	Before Updating (με)	After Updating (με)	Initial Errors (%)	Corrected Errors (%)	Before Updating (με)	After Updating (με)	Initial Errors (%)	Corrected Errors (%)	Before Updating (με)	After Updating (με)	Initial Errors (%)	Corrected Errors (%)
G1_1	−53.1	−51.9	4%	2%	−54.9	−53.7	6%	4%	−55.1	−53.9	5%	3%
G1_2	−28.7	−28.1	6%	3%	−28.1	−27.5	7%	5%	−26.6	−26.0	8%	6%
G1_3	3.3	3.2	9%	6%	7.2	7.0	9%	7%	10.9	10.6	−6%	−8%
G2_1	16.4	16.0	−4%	−6%	21.6	21.1	−4%	−6%	26.1	25.6	−5%	−7%
G2_2	44.8	43.8	6%	4%	52.8	51.7	2%	0%	59.3	58.0	3%	1%
G2_3	61.1	59.8	4%	2%	70.9	69.3	3%	1%	78.5	76.8	4%	2%
G3_1	−116.2	−113.5	4%	1%	−117.4	−114.7	4%	2%	−115.1	−112.4	−3%	−5%
G3_2	−130.2	−127.2	3%	1%	−132.0	−128.9	1%	−1%	−129.8	−126.8	1%	−1%
G3_3	−109.3	−106.8	2%	−1%	−110.3	−107.7	1%	−1%	−107.9	−105.4	1%	−2%
G4_1	126.2	123.3	1%	−1%	134.4	131.2	1%	−1%	138.5	135.3	0%	−2%
G4_2	126.2	123.3	5%	2%	134.4	131.2	3%	0%	138.5	135.3	2%	0%
G4_3	92.4	90.2	4%	2%	99.3	97.0	2%	−1%	103.2	100.7	1%	−1%
G5_1	71.4	69.7	3%	0%	77.7	75.9	4%	1%	81.6	79.7	4%	2%
G5_2	6.9	6.7	14%	12%	10.6	10.3	12%	9%	14.0	13.6	4%	1%
G5_3	−59.5	−58.1	6%	4%	−58.5	−57.1	5%	2%	−55.6	−54.3	5%	2%
*f*(*D*, *t*)	3.2	1.6			2.7	1.3			2.3	1.9		

## Data Availability

The data presented in this study are available on request from the corresponding author.

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
