# Peer review of "Identification of Static Loads in Wharf Mooring Cables Using the Influence Coefficient Method"

_sensors, 2025, doi:10.3390/s25185867_

Round 1
Reviewer 1 Report
Comments and Suggestions for Authors
In the reviewed manuscript, the authors are developing an indirect measurement technique based on strain gauges installed on a steel bollard to assess the forces induced by a mooring cable. For this purpose, preliminary FEM-based simulations together with the optimization of sensor location is performed followed by some experimental studies intended to validate the applicability of the developed approach. While the content of the paper is within the scope of Sensors journal, it could be suggested to revise the paper and address following questions:
1) The authors must very carefully and thoroughly discuss the practical applicability of their approach. Considering Fig. 11 as an example of the deployed system, I could hardly believe that all these cords would survive even during the single ship berthing operation.
2) It is not clear why friction force is neglected in the numerical analysis. Due to wind loads any ship would move back and forth producing sufficient friction between the cable and the bollard.
3) While mentioning in the Introduction "complex marine environmental factors" only very simple experiment is performed in the paper which could be hardly related to real-case loading. With this respect, the authors should very clearly describe why they do believe that such simple case considered in their work might be related to marine applications. Any attempt to shift such research to future work is not acceptable.
Some minor remarks:
1) line 58: What does "changes in frequency" mean here?
2) Subsection 2.1: Certain amount of relevant references should be added when describing ICM method.
3) Line 138: Probably, letter K should be used here instead of F.
4) Line 138: Is such relationship always linear, or some conditions should be met to assume it being linear?
5) Line 188 and further on: it is strongly suggested to check all lower indices - they should look like indices but not second letters within the same line
6) In some cases formulas are given in italic font, in some not. Check this issue.
7) The bollard sketched in Fig. 2 sufficiently diferes from the one shown in Fig. 1. Why there is such strong difference between them?
8) Line 267: Why thin shell elements are chosen? Why not to use solid elements instead?
9) Line 274: Why such load values are considered? Are they related somehow to real-case load caused by ships?
10) Table 1: What was the reason for selecting such population size (N=200)?
11) The authors are suggested to show some figures (at least, one) which illustrate the quality of their GA-solution. Namely, convergence plot is highly desirable.
12) Line 326: it is not clear, why figures 14-16 are referred in the text earlier than figure 10.
13) Fig 12: What does letter "f" mean here? It is not described how exactly this angle "f" is changed in experiments.
14) From Fig. 12 it is not possible to evaluate the discrepancy between experimental strain values and FEM-results.
Author Response
Please see the attachment." in the box if you only upload an attachment.

Reviewer 2 Report
Comments and Suggestions for Authors
The paper presents a novel indirect mooring cable load measurement framework using the Influence Coefficient Matrix (ICM) method combined with Genetic Algorithm (GA)–based strain gauge placement optimization. It models bollard mechanics via finite element analysis, determines optimal sensor locations to improve load identification, and validates the method through simulations and physical experiments. Although the paper shows potential for real-time port safety monitoring, several issues should be explained, validated, calculated, or taken into consideration before publication:
1. Friction between the cable and bollard is assumed negligible, which may not hold true in real ports. In reality, the rope contacts the bollard along an arc, and friction can absorb part of the load before it reaches the bollard’s anchor points. This can result in the bollard experiencing a smaller force than the actual line tension near the vessel, leading to systematic underestimation of identified loads. The friction level varies with rope material, moisture, wrap angle, and vessel motion, making it difficult to model with a fixed correction.
2. Moving the actual cable contact point to the bollard’s axial centre for ICM construction simplifies computation, but may introduce significant errors in field conditions where geometry and contact points vary unpredictably. While computationally convenient, this creates geometric and load-distribution differences that could reduce accuracy, especially when the rope direction changes with vessel movement.
3. Dynamic environmental effects (waves, vessel motion, surge loads) are not modelled. In practice, short-term shocks can cause momentary forces far exceeding static loads, and these may not be correctly captured by a purely static method. This limits applicability to real-time monitoring during storms or heavy port traffic.
4. The rope is flexible and prone to failure under excessive tension, while the bollard is massive and stiff, resulting in very small deformations. Measuring these accurately in a noisy port environment requires high-sensitivity sensors and robust signal conditioning.
5. Experiments were conducted on a small-scale, controlled laboratory setup rather than in a working port environment.
6. Load direction was set manually, introducing potential angular errors that could have been avoided with automated alignment systems.
7. The tested bollard geometry and applied forces were smaller than those of full-scale port installations, and scaling effects were not analysed.
8. While the authors identify three primary sources of error (loading method discrepancy, angle deviation, strain gauge misalignment), they do not quantify their relative contributions.
9. The paper does not provide a sensitivity study to rank which factors (e.g., friction, gauge misplacement, finite element analysis simplification) most strongly affect accuracy.
10. Using exactly five strain gauges (equal to the number of load components) is the theoretical minimum but leaves no redundancy; the benefits of over-sampling are unexplored.
11. The paper does not discuss the “zero drift” phenomenon in strain gauges, which could introduce long-term measurement drift and affect accuracy over extended periods.
Author Response

(The authors gave the same response as above.)

Reviewer 3 Report
Comments and Suggestions for Authors
The paper presents an indirect method for measuring mooring-cable tension at wharves without installing sensors on the rope itself. A bollard-mounted strain-gauge array is optimized by a genetic algorithm, and an influence-coefficient matrix converts the five measured surface strains into the three cable-force components and two moments. Finite-element and scaled-bollard experiments show errors below 10 %, validating the approach for real-time port safety monitoring.
1. The abstract claims the system only needs five gauges, but omits hardware, calibration, and maintenance costs. Please add a short discussion comparing sensor count, installation time, and total cost with quick-release hook sensor systems to help port engineers judge economic viability for large wharf arrays.
2. The tests were conducted in a laboratory with constant temperature and no wind-driven spray. Include a paragraph that reports on or at least simulates the effects of ±20 °C temperature swings, rain, and salt-film accumulation on gauge bonding and signal drift, since these are routine at open quaysides.
3. Real mooring often involves 6–12 ropes sharing one bollard. Either demonstrate how the algorithm can separate simultaneous loads from several cables, or explicitly state that the current method is valid for one cable per bollard and discuss the next modelling steps for multi-rope scenarios.
4. In the literature review part, the review on the data analysis part can be improved. For the experimental monitoring data analysis, there are many machine learning methods, e.g., Bayesian method, Gaussian method, etc., which can refer to the recent publications, e.g., "Data interpretation and forecasting of SHM heteroscedastic measurements under typhoon conditions enabled by an enhanced Hierarchical sparse Bayesian Learning model with high robustness, Towards high-accuracy data modelling", "uncertainty quantification and correlation analysis for SHM measurements during typhoon events using an improved most likely heteroscedastic Gaussian process".
5. In the conclusion part, the future research directions can be described in details.
Author Response

(The authors gave the same response as above.)

Round 2
Reviewer 1 Report
Comments and Suggestions for Authors
The authors, to my mind, have adequately addressed all my questions. The only remark is that although the new title of the paper, i.e., " Identification of Static Loads in Wharf Mooring Cables Using the Influence Coefficient Method" is provided in the coverletter file where the answers to the comments are given, it is not transferred to the revised version of the paper. The authors should check this issue.
Author Response
Thank you for your positive feedback and for bringing this issue to our attention. We sincerely apologize for the inconsistency between the cover letter and the revised manuscript regarding the updated title.
Comments 1: The authors, to my mind, have adequately addressed all my questions. The only remark is that although the new title of the paper, i.e., " Identification of Static Loads in Wharf Mooring Cables Using the Influence Coefficient Method" is provided in the coverletter file where the answers to the comments are given, it is not transferred to the revised version of the paper. The authors should check this issue.
Response 1: The discrepancy occurred because the revised manuscript file with the updated title unfortunately did not upload successfully during our previous submission. We have now verified and re-uploaded the correct version of the manuscript, which includes the revised title: “Identification of Static Loads in Wharf Mooring Cables Using the Influence Coefficient Method”.

Reviewer 3 Report
Comments and Suggestions for Authors
The author conducted a large number of experiments, but the data analysis of the experiments was not sufficient. It is recommended to deeply explore the experimental results, fully explore the patterns and possible causes revealed in the experiments.
The literature review part can be improved by including more recent literature on load sensing technology, especially the RFID passive sensing technology, which is missing in the current review section, for instance the literature as follows, Towards long-transmission-distance and semi-active wireless strain sensing enabled by dual-interrogation-mode RFID technology, A Concise State-of-the-Art Review of Crack Monitoring Enabled by RFID Technology,
Author Response
We are deeply grateful for the thorough and constructive feedback provided by the reviewer. We sincerely appreciate the time and effort dedicated to improving our manuscript. In response to the comments, we have carefully revised the manuscript to address all the raised points.
Comments 1: The author conducted a large number of experiments, but the data analysis of the experiments was not sufficient. It is recommended to deeply explore the experimental results, fully explore the patterns and possible causes revealed in the experiments.
Response 1: We fully agree that a more thorough exploration of the underlying patterns and causes would significantly strengthen the paper. Accordingly, we have carefully re-examined our experimental results and substantially revised the manuscript to provide a more in-depth discussion of the observed trends and their potential reasons. Major amendments include:
Lines 472-475 :”[In these figures, load identification values from finite element simulation and experiment are presented as two sets of scatter points, with their corresponding deviations from the equality lines indicated by dashed error bars.]”
Lines 496-531 :”[Analysis of the experimental results revealed the following patterns:
1) Identification of load components. As shown in Fig. 15, the maximum errors associated with the five load components varied. The identification accuracy for the horizontal loads (Fx and Fy) was higher than that for the vertical load (Fz)and the bending moments (Mx and My). These discrepancies can be primarily attributed to differences in load application methods. In the finite element analysis, forces and equivalent moments were idealized as applied at the axial center of the bollard. During practical experiments, however, the load was introduced eccentrically at the edge, leading to variations in structural strain responses. Since the horizontal load components act through the top center, the strain response differences between the two loading modes are relatively small. In contrast, vertical loads and the moments induced by them elicit more pronounced changes in strain response. Since the location and direction of the mooring cable load are unknown prior to identification, the unit load in FEA can only be applied using the center loading method when constructing the influence coefficient matrix. Experimentally, however, structural strains were measured under edge-loading conditions. This inconsistency in loading configuration contributes to the observed identification errors.
2) Identification of load direction. As illustrated in Fig. 16, the identification errors for both the horizontal angle (θ) and vertical angle (Φ) were within ±10%. The errors did not exhibit a consistent trend and were largely stochastic across the tested cases. This randomness may be attributed to the two main causes: deviations in load direction and measurement inaccuracies in the strain gauges. Although the tension meter accurately controlled the magnitude of the cable load during testing, the direction was not mechanically regulated and relied on manual alignment, introducing directional deviation. Consequently, the identified load direction and its components also deviate from the required values. The angular setting errors can be significantly reduced through automatic alignment systems. Additionally, strain gauges inherently possess measurement errors. Positioning strain gauges accurately on a cylindrical bollard was challenging, often resulting in misalignment between the actual position and the intended measurement points. Despite implementing finite element updating to mitigate these errors, complete elimination was not feasible.
3) Identification of mooring cable load. Although the maximum errors in the identification of individual load components ranged from 10% to 20%, the error in the identified mooring cable load (F) did not exceed 5%. This can be attributed to the fact that the cable tension is a result value derived from multiple components (see Eq. (18)), and the maximum errors of each component did not occur at the same time.]”
Lines 552-554 :”[Error analysis reveals that the maximum relative error in load magnitude and direction remains below 10%, which validates the accuracy and reliability of the proposed method.]”
Comments 2: The literature review part can be improved by including more recent literature on load sensing technology, especially the RFID passive sensing technology, which is missing in the current review section, for instance the literature as follows, Towards long-transmission-distance and semi-active wireless strain sensing enabled by dual-interrogation-mode RFID technology, A Concise State-of-the-Art Review of Crack Monitoring Enabled by RFID Technology,
Response 2: We sincerely thank the reviewer for this excellent suggestion. The literature on load sensing technology, especially the RFID passive sensing technology has been supplemented in the literature review section.
Line 71-75 :”[The evolution of sensing technology further supports this method; for instance, emerging passive wireless sensing techniques like Radio-Frequency Identification (RFID) offer promising solutions for long-term strain monitoring with advantages of no external power requirement and ease of installation [13-15].]”
